# Effects of Environmentally Relevant Concentrations of Antipsychotic Drugs (Sulpiride and Clozapine) on Serotonergic and Dopaminergic Neurotransmitter Systems in Octopus Brain Tissue

**Xijian Peng [1,2], Qiuxia Xu [1,2], Yuanming Guo [1,2] and Bo Zhang [3,\***

1   Marine and Fishery Institute of Zhejiang Ocean University, Zhoushan 316021, China
2   Key Laboratory of Sustainable Utilization of Technology Research for Fishery Resource of Zhejiang Province, Marine Fisheries Research Institute of Zhejiang, Zhoushan 316021, China
3   The Key Laboratory of Environmental Pollution Monitoring and Disease Control, Ministry of Education, School of Public Health, Guizhou Medical University, Guiyang 550025, China
\*   Correspondence: sky_zhangbo_sea@163.com

**Abstract:** Pharmaceuticals and personal care products (PPCPs) from specific activities often enter surface and groundwater, adversely affecting the physiological functions of non-target organisms and posing a serious threat to a wide range of aquatic organisms. Therefore, the aim of this study was to investigate the effects of environmentally relevant concentrations of the antipsychotic drugs sulpiride and clozapine on dopaminergic (DAergic) and serotonergic (5-hydroxytryptaminergic, i.e., 5-HTergic) neurotransmitter systems in the brain of a short-arm octopus (*Octopus ochellatus*). *Octopus ochellatus* adults were exposed to environmentally relevant concentrations of sulpiride, clozapine, or a mixture of sulpiride and clozapine. The effects of the drug on the transcription and expression levels of major functional molecules in the DAergic and 5-HTergic systems of the brain were analyzed. By antagonizing the dopamine receptor D2 (DRD2) or 5-hydroxytryptamine receptor 2A (5-HTR2A), the two drugs induced abnormal transcription and expression levels of important functional molecules in the brain's DA and 5-HT signaling pathways. In addition, dose-dependent adverse reactions were observed with these antipsychotics. Our results suggest that sulpiride and clozapine interfere with DAergic and 5-HTergic neurotransmitter systems in the brain of Amphioctopus fangsiao (*O. ochellatus*), possibly affecting brain functions, such as reproduction, predation, camouflage, learning, and memory. As a result, they pose a serious threat to the health of Amphioctopus fangsiao.

**Keywords:** PPCPs; Amphioctopus fangsiao; brain; DAergic; 5-HTergic



## 1. Introduction

With the increase in the production and consumption levels of pharmaceuticals and personal care products (PPCPs), environmental pollution caused by PPCPs has become an emerging global public health problem. Among them, the issue of drugs entering environmental water bodies at a low concentration level for a long time is a serious concern [1–4]. Drugs can interfere with the normal physiological functions of the organisms expressing the non-drug target, seriously threatening their health [5]. Therefore, studies on the effects of drug pollution on aquatic organisms have become a research hotspot in ecotoxicology.

Schizophrenia, a mental disorder, refers to a group of severe mental disorders of unknown etiology. The basic pathological manifestations of schizophrenia are hyperfunction of the mesolimbic and mesocortical pathways of the dopamine (DA) system accompanied by a 5-hydroxytryptamine (5-HT) metabolism disorder [6]. The first-generation antipsychotics, which are $D_2$ receptor antagonists, mainly act on the central nervous system's dopamine $D_2$ receptor [7]. Second-generation antipsychotic drugs mainly act on 5-HT$_2$ receptors in the central nervous system, with high antagonism to 5-HT$_2$ receptors and relatively low antagonism to $D_2$ receptors [8]. Sulpiride (a first-generation benzodiazepine

antipsychotic) and clozapine (a second-generation benzodiazepine antipsychotic) are representative antipsychotic drugs that are widely used because of their good efficacy and low side effects.

Through a variety of pathways such as discharges of wastewater effluents, pharmaceuticals and their derivatives can enter aquatic systems exposed to aquatic organisms [9]. Evidence suggests that sulpiride and clozapine are widely distributed in water bodies, but they have not been reported in coastal and seawater environments. A study found that the concentration of clozapine in the effluent and sludge of a sewage treatment plant in the Guangdong Province of China was approximately 44.5 ng/L and 62.9 ng/g, respectively [10]. Similarly, an analysis of the water from ten sewage treatment plants in central and southern Germany revealed the presence of 111–1100 ng/L and 110–338 ng/L sulpiride in the inflow and outflow waters, respectively [11]. Furthermore, the data from psychiatric hospitals are alarming. The effluent water of the sewage treatment system of a psychiatric hospital in Beijing, China had 432–10,833 ng/L sulpiride and 295–8183 ng/L clozapine [12].

Cephalopods have a complex brain structure with cartilage around it; a highly concentrated nervous system; and excellent motor, sensory, and cognitive abilities. Amphioctopus fangsiao is an important neurophysiological species extensively distributed in coastal China [13]. The nervous system of an octopus, including the circumesophageal brain, a pair of optic lobes, and the axial nerve cords on each arm, varies greatly in size and organization compared to other mollusks [14,15]. In addition, it has been found that the formation of long-term memory in an octopus depends on the vertical lobe and the long-term potentiation (LTP) in this structure [16]. Cuttlefish, which are also cephalopods, have the ability to capture visual and chemical signals from predators and visual signal features of prey in the environment through learning in their infancy [17].

DA, an important neurotransmitter in the cephalopod nervous system, is widely found in all parts of the cephalopod nervous system, except the fin lobe and upper esophageal lobe. DA is unevenly distributed in all brain regions of cephalopods [18]. In addition, DA in the cephalopod brain is very sensitive to reserpine and pargyline [19,20]. DA plays an important role in regulating brain functions related to an octopus's movement, motivation, behavior, learning, and memory [21,22]. Studies have shown that injecting DA (or its agonists) into the head of the octopus can cause a variety of abnormal movements in the octopus's mantle and darken the body surface [23]. DA can also affect invertebrate reproduction. It has been shown that the great significance of DA and its receptors in the early development of oyster ovar [24] 5-HT was first found in the serum and is therefore known as serum tensin. Furthermore, 5-HT is widely distributed but is low in cephalopods and is only enriched in a few areas, such as the middle shell slit and frontal cortex [18]. In addition, 5-HT, a neurochemical messenger, promotes the production of activation-dependent LTP and plays an important role in octopus memory formation by enhancing the signal transmission between neurons involved in memory formation [25,26]. Some scholars have found that 5-HT injected into the octopus by the aortic perfusion technique can cause clear motor, including ink-jet and excretion [23]. The source and distribution of most 5-HT neurons in cephalopods are similar to those of DA and may assist DA in regulating neuronal activity [27–29]. Moreover, 5-HT and its receptors play a role in regulating the survival and reproduction of aquatic mollusks. Indeed, 5-HT can regulate the feeding movement [30], retraction reflex [31], and defense [32] of sea hares, as well as gametogenesis, embryo development, and the induction of oocyte maturation and ovulation in the Pacific oyster [33].

In the literature, a large amount of ecotoxicological data on antipsychotic drugs are present. Unfortunately, there are little data on antipsychotic drugs as environmental contaminants and their effects on aquatic species, especially cephalopods. The objectives of this study were to investigate the effects of antipsychotic drugs on the physiological functions of the octopus, and to provide basic data for further study on the effects of antipsychotic drugs on the health of aquatic organisms.

## 2. Materials and Methods

### 2.1. Drugs

Sulpiride (S8010, CAS: 15676-16-1) and clozapine (C6305, CAS:5786-21-0) were purchased from Sigma-Aldrich Co., LLC (Shanghai, China). Sulpiride and clozapine were dissolved in a 1% ethanol solution to prepare a final concentration of 1 mg/L drug solution, which was stored at $23 \pm 1\ °C$ to avoid light for use.

### 2.2. Experimental Animals and Disposal Methods

All octopuses ($n = 100$, half female and half male) were collected from the Zhoushan area, Zhejiang Province, China. Octopuses were randomly divided into ten tanks ($60 \times 30 \times 40$ cm). Each closed-loop-system tank was supplied with 40 L artificial seawater prepared at a salinity of 29 $g/L^{-1}$ (Tropic Marin, Wartenberg, Germany) and aerated until the end of the experiment. The light/dark cycle was 14:10 h, and the temperature was maintained at $16 \pm 1\ °C$. The octopuses were fed white shrimp twice a day. After acclimatization for 3 days, the octopuses were randomly divided into nine treatment groups (half female and half male) and a control group of 10 octopuses per group. Sulpiride and clozapine concentrations were chosen to approximately measure WWTP effluents concentrations according to previous studies [10,34], calculated considering the average concentration of different antipsychotic drugs in China's wastewater and coastal waters [35,36]. Two higher concentrations of 50 and 100 ng/L were also tested to mimic and compare simultaneous exposure to multiple benzodiazepines occurring in nature. The same volume of artificial seawater (containing 100 microliters of 1% ethanol) was added to the control group. The solutions were replaced every 24 h to ensure uniform concentration of the drug. The quality of water was checked every day. Then, octopuses were sacrificed, and their brain tissues were collected quickly after 14 days of exposure. There are no specific ethical regulations relating to experimental work with octopuses in China. All animal studies were conformed to Directive 2010/63/EU because the directive emphasizes the protection of cephalopods [37,38].

### 2.3. RNA Extraction and qPCR

The animals were anesthetized using a seawater solution with $MgCl_2$ [39] to remove all the brain tissue and extract RNA. Total RNA was extracted using the RNAeasy Animal RNA Extraction Kit (Biyuntian, Shanghai, China) according to the manufacturer's instructions. Each sample was treated with a Turbo DNase Kit (Ambion Inc., TX, USA) to degrade the contaminating DNA, according to the manufacturer's instructions. For all RNA samples, DNA contamination was detected using PCR with β-actin primers. The absorbance at 260 nm and the ratios of 260/280 nm and 260/230 nm were measured using a Nanodrop (NanoDrop 2000C Spectrophotometer; Thermo Scientific, MA, USA) to determine the purity and quality of the extracted RNA. The cDNA was stored at $-20\ °C$, and the relative expression levels of each receptor were calculated using the $2^{-\Delta\Delta CT}$ method.

To understand the effects of drugs on the normal functions of DA and 5-HT neurotransmitter systems in the octopus brain tissue, we selected important functional molecules, including monoamine oxidase (MAO), and the DA transporter (DAT) and DA receptors (DRD1, DRD2, DRD3) of the DA neurotransmitter system, and the 5-HT transporter (SERT) and 5-HT receptors (5-HTR1, 5-HTR1B, 5-HTR2A) of the 5-HT neurotransmitter system. The primer sequences used are listed in Table 1. The qPCR amplification was performed using the CFX Connect Detection System (Bio-Rad, California, USA) and BeyoFast SYBR Green qPCR Mix (Biyuntian, Shanghai, China) with three technical and biological replicates. The PCR conditions were as follows: 2 min at 95 °C and 40 cycles at 95 °C for 15 s, and 30 s at 60 °C. The RT-PCR technique was adapted from a previous study [40]. Primer specificity was determined using a melting curve analysis. The expression data were normalized to a β-actin expression. The $2^{-\Delta\Delta CT}$ method was used to calculate the relative quantification of the mRNA levels of each gene in each group.

**Table 1.** Primer sequences.

| Genes | Primer Type | Sequences |
| --- | --- | --- |
| DRD1 | Forward | CGTCGTCGTCGTTGTCGTCATC |
| | Reverse | AGGCAGGCAAGCTGTTGATGTG |
| DRD2 | Forward | GAAGGTGCGTAGAGATGCGAAGAC |
| | Reverse | CCCTCCAAACTCACTCCACTCATTC |
| DRD3 | Forward | TGTAGGAGAGTCACCACGAAGGC |
| | Reverse | ACTACCAGGGAATCGGTTGTCATTG |
| DAT | Forward | TGATTGGTTGGTGCGTTGCTCTC |
| | Reverse | GCTTGCTTGTGGTTAGTTGGATGTG |
| 5-HTR1 | Forward | CGGAAGGCTGCCAGAGTGATTG |
| | Reverse | TCTAAAGAGGCACGAGCGAAATTGG |
| 5-HTR1B | Forward | CGGAAGGCTGCCAGAGTGATTG |
| | Reverse | TCCGATCAAGGCAACCATGAAGAAG |
| 5-HTR2A | Forward | CATCTCCTGCGAGTCCAAGACATTC |
| | Reverse | TTAGTAGCCTTCCGTTCCGTTTGC |
| SERT | Forward | AGTCTGGATAACAGCCACCGTACC |
| | Reverse | TCCCATCACTGGAACCAGGAAGAG |
| MAO | Forward | GCAACCCAATCGCCTTCAAATCAC |
| | Reverse | CTGTCTTCCGAGTGTCCATGTTCC |
| β-actin | Forward | CCCAGGTATTGCTGACCGTATGC |
| | Reverse | GAAGGTGGACAGAGAAGCCAAGATG |

## 2.4. Western Blot Analysis

The whole brain was collected after anesthesia, and the vertical complex was extracted. Total protein was extracted using a commercial kit (Beyotime Institute of Biotechnology, Shanghai, China). A Western blot analysis was performed as previously described derivative [41]. Briefly, the tissue blocks were rinsed 2–3 times with pre-cooled phosphate buffered saline, and lysate (10 times the tissue volume) was added (the protease inhibitor was added within a few minutes before use), and a homogenization program was set up. After homogenization, the homogenate tube was placed on ice for 30 min and shaken every 5 min to ensure complete tissue disruption. Subsequently, the homogenate was centrifuged at 12,000 rpm for 10 min at 4 °C, and the supernatant was collected to form the total protein solution. Protein concentrations were determined using a BCA protein assay kit (Beyotime Institute of Biotechnology, Shanghai, China). Equal amounts of protein (30 μg) were separated on 10% or 12% sodium dodecyl sulfate-polyacrylamide gels and transferred to polyvinylidene difluoride membranes. The membrane was sealed with 1% bovine serum albumin for 1 h and then treated with rabbit anti-5HTR2B (dilution, 1:700; Solarbio Science & Technology, Beijing, China), rabbit anti-5HTR1B (dilution, 1:500; Thermo Fisher Scientific, MA, USA), rabbit anti-5HTR1 (dilution, 1:1000; Abcam Corporation, Cambridge, UK), rat anti-DAT (dilution, 1:500; Santa Corporation, Shanghai, China), rabbit anti-SERT (dilution, 1:1000; Abcam Corporation, Cambridge, UK), or mouse anti-rat β-actin (dilution, 1:500; ImmunoWay Biotechnology, Newark, DE, USA) overnight at 4 °C, and then incubated with alkaline phosphatase goat anti-rabbit IgG (dilution, 1:1000; ZSGB-BIO, Beijing, China) for 1 h. Western Blue® stabilized substrate alkaline phosphatase (Promega Corporation, Madison, WI, USA) was used to detect immune response signals, and then the membrane was air-dried and imaged using an image analyzer (Bio-Rad Laboratories, Inc., Hercules, CA, USA). The band intensities were tested using Image J v1.48 software (National Institutes of Health, MIH, Bethesda, USA). The Western blot analysis was repeated three times.

## 2.5. Statistical Analysis

The results are presented as the mean ± standard error. The relative transcription differences of DAT, DRD1, DRD2, DRD3, SERT, 5-HTR1, 5-HTR1B, and 5-HTR2A, and the band intensities of DAT, 5-HTR1, 5-HTR1B, and SERT among different groups were analyzed by a two-way repeated measures analysis of variance, followed by the LSD multiple comparison test. A *p*-value < 0.05 was considered statistically significant.

## 3. Results

### 3.1. Effects of Sulpiride and Clozapine on the Transcription of Functional Molecules of DA and 5-HT Neurotransmitter Systems in Octopus Brain Tissue

We employed RT-PCR to assess the mRNA transcription levels. Sulpiride induced the abnormal transcription of DRD1, DRD2, DRD3, DAT, 5-HTR1, 5-HTR1B, 5-HTR2A, SERT, and MAO in octopus brain tissues after 14 days of continuous exposure ($p < 0.0001$). Among them, DRD1, DRD3, 5-HTR1, 5-HTR1B, 5-HTR2A, and MAO were upregulated in the 20 ng/L exposure group ($p < 0.05$). Meanwhile, DRD1, DRD2, DAT, 5-HTR1, 5-HTR1B, 5-HTR2A, SERT, and MAO were downregulated in the 50 ng/L exposure group ($p < 0.05$). Furthermore, DRD1, DAT, and SERT were downregulated in the 100 ng/L exposure group ($p < 0.05$). In addition, sulpiride induced the downregulation of DRD2 transcription in the 50 ng/L exposure group ($p < 0.0001$) (Figure 1). The mRNA level of DRD2 was significantly decreased in the sulpiride group, but only in the 50 ng/L ($p$ values 0.0386).

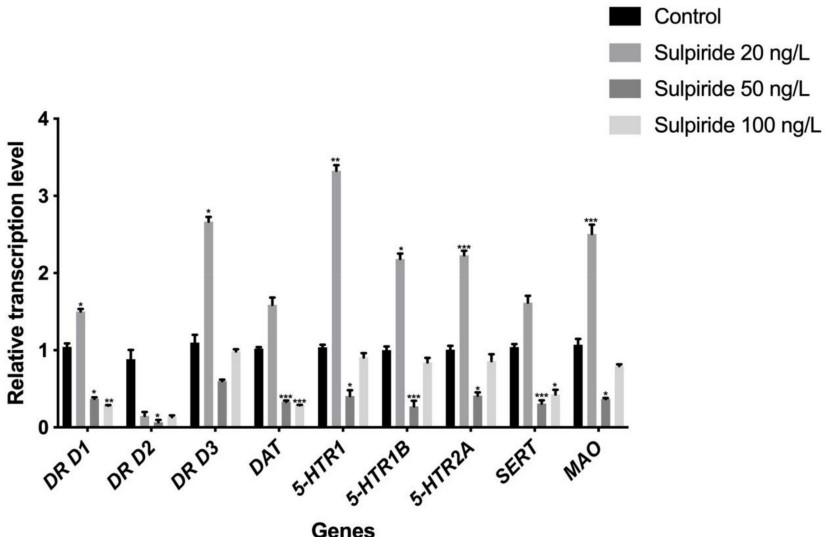

**Figure 1.** Sulpiride exposure for 14 days induced changes in gene transcription levels of functional molecules of DA and 5-HT neurotransmitter systems in octopus brain tissue. Note: * compared with the control group, $p < 0.05$; ** compared with the control group, $p < 0.01$; *** compared with the control group, $p < 0.001$; $n = 5$.

Clozapine induced the abnormal transcription of DRD1, DRD2, DRD3, DAT, SERT, MAO, 5-HTR1, 5-HTR1B, and 5-HTR2A in octopus brain tissues after 14 days of continuous exposure ($p < 0.0001$). Specifically, DRD1, DAT, 5-HTR2A, and SERT were downregulated in the 20 ng/L exposure group ($p < 0.05$). In the 50 ng/L exposure group, DRD1, DRD3, and DAT were significantly decreased ($p < 0.05$), while 5-HTR1 and 5-HTR1B were upregulated ($p > 0.05$). In the 100 ng/L clozapine exposure group, DRD1 ($p < 0.05$), DAT ($p < 0.001$), and MAO ($p < 0.05$) were downregulated (Figure 2).

Sulpiride and clozapine induced the abnormal transcription of 5-HTR1, 5-HTR1B, 5-HTR2A, and SERT in octopus brain tissues after 14 days of continuous exposure ($p < 0.0001$). The expression levels of DRD1, 5-HTR1, 5-HTR1B, 5-HTR2A, SERT, and MAO were upregulated in the 20 ng/L mixed exposure group ($p < 0.001$); DRD3 and DAT were upregulated in the 20 ng/L mixed exposure group ($p < 0.05$). Furthermore, DRD1, DRD3, DAT, 5-HTR1, 5-HTR1B, 5-HTR2A, SERT, and MAO were upregulated ($p < 0.01$) and DRD2 was downregulated ($p > 0.05$) after 50 ng/L mixed exposure. The transcription levels of DRD1 ($p < 0.05$) were downregulated, but 5-HT1, 5-HT1B, and MAO ($p < 0.01$) were upregulated after 100 ng/L sulpiride and clozapine mixed exposure (Figure 3).

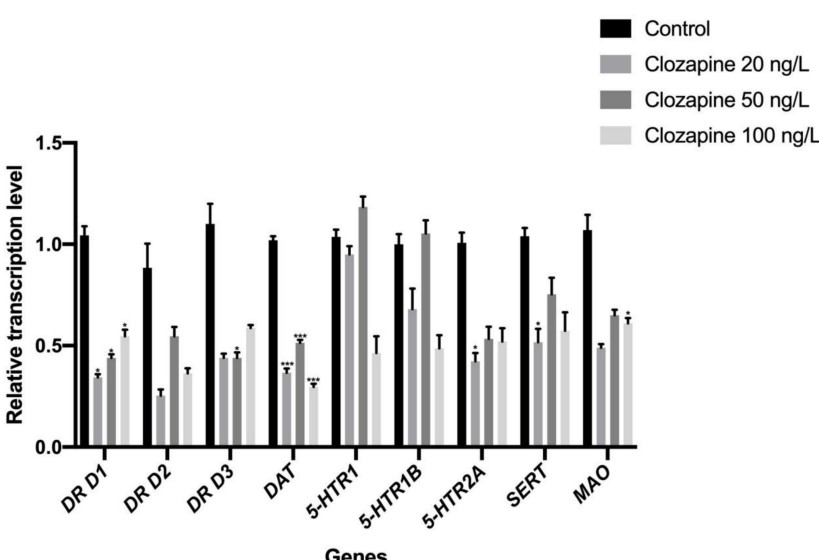

**Figure 2.** Clozapine exposure for 14 days induced changes in gene transcription levels of functional molecules of DA and 5-HT neurotransmitter system in octopus brain tissue. Note: * compared with the control group, $p < 0.05$; *** compared with the control group, $p < 0.001$; $n = 5$.

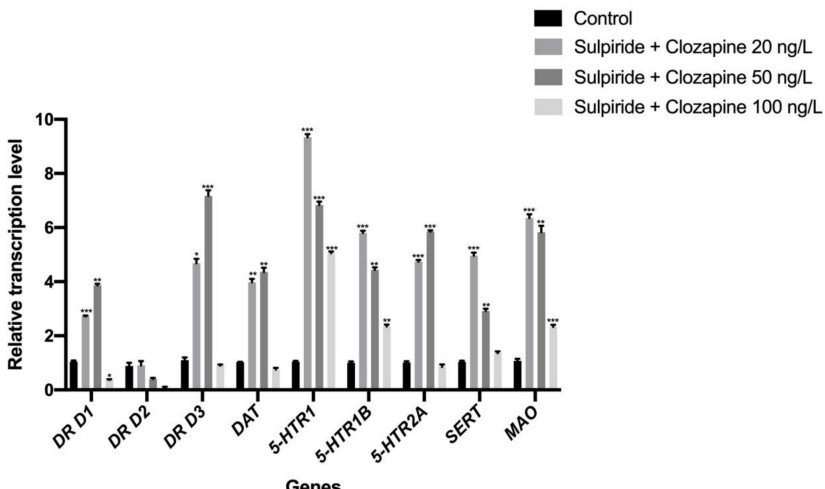

**Figure 3.** Sulpiride and clozapine mixed exposure for 14 days induced changes in the expression levels of functional molecules of DA and 5-HT neurotransmitter systems in octopus brain tissue. Note: * compared with the control group, $p < 0.05$; ** compared with the control group, $p < 0.01$; *** compared with the control group, $p < 0.001$; $n = 5$.

*3.2. Effects of Sulpiride and Clozapine on Protein Expression Levels of Functional Molecules of DA and 5-HT Neurotransmitter System in Octopus Brain Tissue*

The changes of protein levels in octopus brain tissue are shown in Figure 4.

In addition, we also assessed the protein levels of DAT, 5-HTR1, 5-HTR1B, 5-HTR2B, and SERT. The results showed that 5-HTR1, 5-HTR1B, and SERT increased after sulpiride stimulation in octopus brain tissues ($p < 0.0001$), while no significant effect was seen for the concentration of 20 and 50 ng/L of sulpiride on protein levels of DAT and 5-HT2B compared with the control group. The expressions of 5-HT1B and SERT were upregulated in the 20 ng/L sulpiride group ($p > 0.05$). The protein expression of 5-HTR1 and SERT was significantly induced by 50 ng/L sulpiride exposure compared with the control ($p < 0.05$). Furthermore, 5-HTR1B and SERT expressions were increased significantly in the 100 ng/L sulpiride exposure group ($p < 0.05$) (Figure 5).

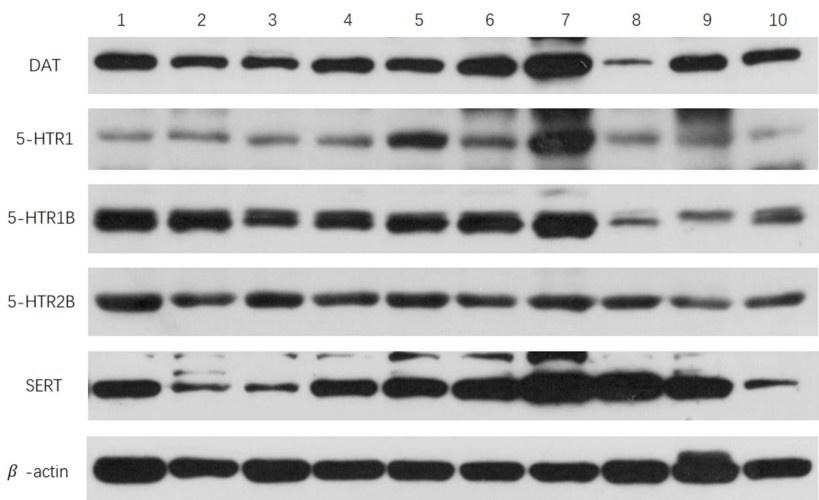

**Figure 4.** Effects of sulpiride and clozapine on protein expression levels of functional molecules of DA and 5-HT neurotransmitter systems in octopus brain tissue. Note: 20 ng/L clozapine exposure group; 2. 50 ng/L clozapine exposure group; 3. 100 ng/L clozapine exposure group; 4. 20 ng/L sulpiride exposure group; 5. 50 ng/L sulpiride exposure group; 6. 100 ng/L sulpiride exposure group; 7. 20 ng/L (sulpiride + clozapine) mixed exposure group; 8. 50 ng/L (sulpiride + clozapine) mixed exposure group; 9. 100 ng/L (sulpiride + clozapine) mixed exposure group; 10. control group.

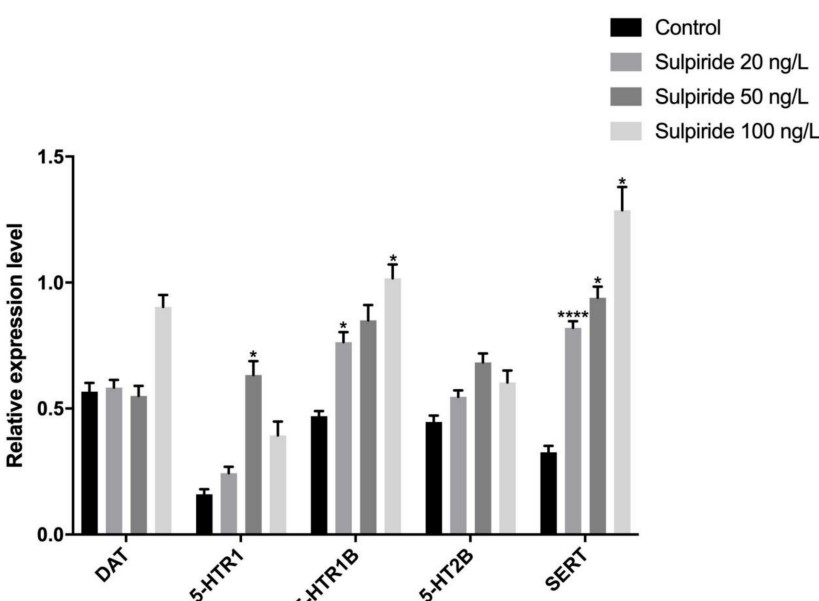

**Figure 5.** Sulpiride exposure for 14 days induced changes in protein expression levels of functional molecules of DA and 5-HT neurotransmitter systems in octopus brain tissue. Note: * compared with the control group, $p < 0.05$; **** compared with the control group, $p < 0.0001$; $n = 5$.

DAT, 5-HTR1, 5-HTR1B, 5-HTR2B, and SERT expression levels were abnormal in the octopus brain tissues after 14 days of clozapine exposure ($p < 0.05$). The expressions of DAT, 5-HTR1B, 5-HTR2B, and SERT were upregulated in the 20 ng/L clozapine exposure group ($p > 0.05$). The expressions of 5-HTR1, 5-HTR2B, and SERT were upregulated ($p > 0.05$) and the expression of 5-HTR1B was upregulated ($p < 0.05$) in the 50 ng/L group. DAT expression was decrease significantly ($p < 0.05$) after the 100 ng/L treatment (Figure 6).

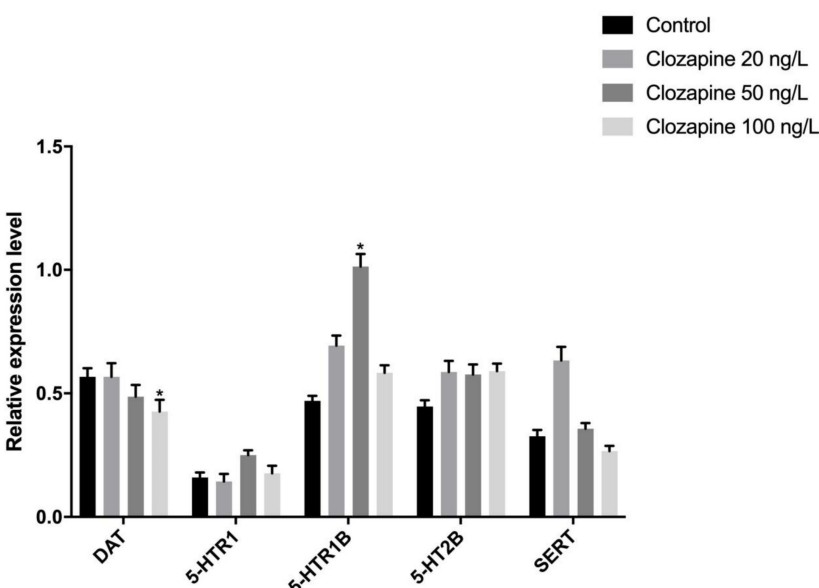

**Figure 6.** Clozapine exposure for 14 days induced changes in protein expression levels of functional molecules of DA and 5-HT neurotransmitter systems in octopus brain tissue. Note: * compared with the control group, $p < 0.05$; $n = 5$.

As shown in Figure 7, abnormal expression levels of DAT, 5-HTR1, 5-HTR1B, 5-HTR2B, and SERT in octopus brain tissues were induced by sulpiride and clozapine mixed exposure for 14 days ($p < 0.05$). In addition, 5-HTR1 and SERT expressions were upregulated ($p < 0.0001$); DAT and 5-HTR1B expressions were upregulated in the 20 ng/L group ($p < 0.05$). The expression of SERT in the 50 and 100 ng/L sulpiride and clozapine mixed exposure group increased significantly ($p < 0.05$). In addition, the expression levels of 5-HTR1B, 5-HTR2B, and SERT in the 20 ng/L sulpiride and clozapine mixed exposure group showed a dose-dependent decrease compared to the 50 ng/L group.

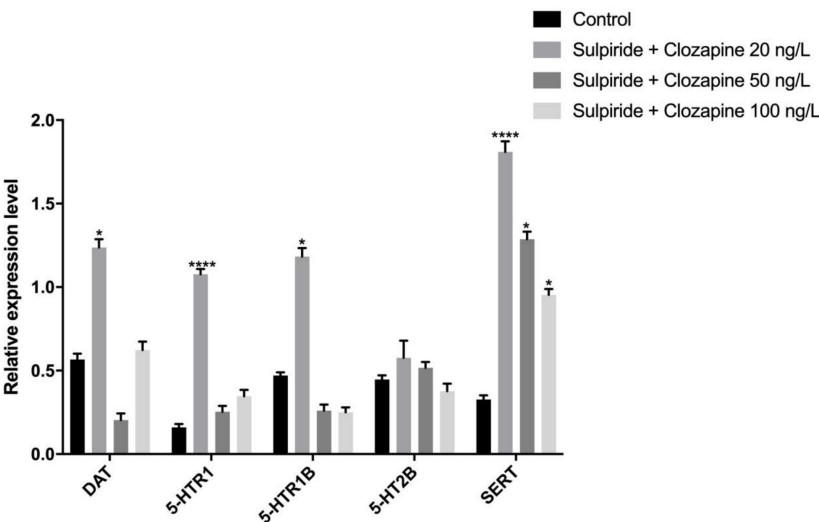

**Figure 7.** Sulpiride and clozapine mixed exposure for 14 days induced changes in protein expression levels of functional molecules of DA and 5-HT neurotransmitter systems in octopus brain tissue. Note: * compared with the control group, $p < 0.05$; **** compared with the control group, $p < 0.0001$; $n = 5$.

## 4. Discussion

Drugs are the most common pollutants in aquatic environments because of their physiological activity and resistance to degradation and may cause long-term harm to aquatic ecosystems [42]. Antipsychotics are the most prescribed active drugs worldwide, and the

COVID-19 pandemic that began in late 2019 and continues to this day has further increased their usage [43–45]. Studies have shown that psychotropic substances are often detected in sewage, surface water, and groundwater [46,47], and wastewater treated by sewage treatment plants can adversely affect aquatic life, even at very low concentrations [48].

In this study, sulpiride and clozapine were selected as representative antipsychotic drugs to investigate their effects on DA and 5-HT neurotransmitter systems in the brain tissue of an octopus. Sulpiride has a high affinity for the DRD2 receptor and can exert its pharmacological effect by antagonizing DRD2 [49]. Simultaneously, there is a certain level of interaction between DRD1 and DRD2 receptors; DRD1 does not play a direct role, but its existence creates conditions for the DRD2 to perform its corresponding functions. The results showed that sulpiride exposure induced the downregulation of DRD1 and DRD2 in the octopus brain tissue. The transcriptional levels of DAT, DRD1, DRD3, and MAO were upregulated in the brain tissues of octopuses exposed to sulpiride at 20 ng/L. A possible cause is that sulpiride antagonizes the DRD2 receptor, upregulating the transcription levels of DAT, DRD1, and DRD3. Low-dose drug exposure causes compensatory effects in the octopus brain tissues. Signal transduction of the DA and 5-HT neurotransmitter systems can be enhanced by promoting 5-HT recovery in the synaptic cleft and 5-HT receptor transcription and expression. The transcription levels of functional molecules of the DA and 5-HT systems were downregulated. Meanwhile, 50 ng/L sulpiride exposure had a greater effect on the transcription levels of functional molecules of the DA and 5-HT systems than 100 ng/L sulpiride exposure. The results showed that the transcription levels of all functional molecules in the 50 ng/L sulpiride group were lower than those in the control group, and the degree of downregulation was more significant than that in the 100 ng/L sulpiride exposure group, except for DRD1. The reason for this phenomenon may be that a higher dose of sulpiride antagonizes DRD2 in the pre- and post-synaptic membranes and has a negative feedback regulatory effect on the activity of the DA neurotransmitter system, leading to a decrease in the level of DA in the synaptic space. In contrast, sulpiride induced the dose-dependent upregulation of functional proteins of the DA and 5-HT neurotransmitter systems in the octopus brain tissue. This phenomenon may be caused by early adverse reactions to the drugs in the body. Under these conditions, many functional protein molecules have already been expressed and play a regulatory role in the early stage of drug exposure; therefore, the upregulation of protein expression levels is accompanied by the downregulation of transcription levels.

Clozapine antagonizes both 5-HT and DA receptors. Among them, clozapine has the strongest affinity for the serotonin receptor 5-HTR2A (Ki = 2.5 nm) and a weak affinity for the dopamine receptors DRD1 (Ki = 254 nm), DRD2 (Ki = 172 nm), and DRD3 (Ki = 555 nm) [50]. The 5-HTR2A receptor is involved in the regulation of neuronal activity and affects 5-HT release through negative feedback [51]. Simultaneously, the 5-HTR2A receptor expressed in the membrane of DA neurons can regulate the tonic release of DA, leading to phasic activation of DA neurons. In addition, the activation of 5-HTR1 receptors in the pre-synaptic membrane inhibits neuronal activity and leads to decreased transmitter release (including 5-HTR1A, which regulates neuronal firing, and 5-HTR1B, which regulates 5-HT release) [52]. The 5-HT receptors may control the activity of DA neurons in a state- and region-dependent manner [53]. Therefore, clozapine antagonizes 5-HTR2A, DRD1, DRD2, and DRD3 in the octopus brain tissue, resulting in decreased DA synthesis, decreased signaling activity of the DA neurotransmitter system, downregulation of the transcription of functional molecules, and increased protein expression levels of the 5-HT neurotransmitter system. Specifically, the transcription levels of DAT, DRD1, DRD2, DRD3, DAT, 5-HTR2A, SERT, and MAO were downregulated, and the protein expression levels of 5-HTR1, 5-HTR1B, 5-HTR2B, and SERT were upregulated.

Because sulpiride and clozapine, having similar pharmacological effects, are representative antipsychotic drugs, in order to explore whether they have synergistic effects, a co-treatment experiment was designed in this study. The results showed that there was no synergistic effect, and sulpiride combined with clozapine at 20 ng/L increased the

activity of the DA neurotransmitter system by increasing DA synthesis, transport, and upregulation of DA receptor levels, which was accompanied by increased 5-HT system activity manifested by the upregulation of the transcription levels of DRD1, DRD2, DRD3, DAT, 5-HTR1, 5-HTR1B, 5-HTR2A, SERT, MAO, and the protein expression levels of DAT, 5-HTR1, 5-HTR1B, 5-HTR2B, and SERT. The 50 ng/L mixed drug exposure group was similar to the 20 ng/L mixed drug exposure group, but the difference was that the transcription of DA neurotransmitter system-related receptors was more significantly upregulated in the 50 ng/L group than in the 20 ng/L mixed drug exposure group. In contrast, the transcription and expression of 5-HT receptors in the 50 ng/L group were weaker than those in the 20 ng/L drug mixture exposure group. The reason may be the existence of a "low-dose effect", wherein a low dose (or concentration) would cause an effect that a higher dose/concentration would not [54]. In addition, 100 ng/L mixed drug exposure resulted in decreased activity of the DA and 5-HT neurotransmitter systems, which manifested through the downregulated transcription levels of DRD3, DAT, 5-HTR2A, and SERT, and the downregulated protein expression levels of DAT, 5-HTR1, 5-HTR1B, and 5-HTR2B. It is worth noting that the DAT transcription and protein expression levels were inconsistent in the 50 ng/L and 100 ng/L mixed drug exposure groups, showing that the DAT transcription level was upregulated, but the protein expression level was downregulated. In the 100 ng/L mixed drug exposure group, the DAT transcription level was downregulated, but the protein expression level was upregulated. A possible reason is that the effects of high concentrations of mixed drugs on the body appear earlier and induce the upregulation of a large number of functional protein molecules in the early exposure stage.

Previous studies have shown that exposure to antipsychotic drugs can cause abnormal levels of neurotransmitters in aquatic organisms, interfere with their normal physiological functions, and lead to abnormal behaviors [55–59]. However, no similar abnormal behavior has been observed in this study. Among them, the dopaminergic (DAergic) system can affect the movement, cognition, reward, feeding behavior, and social behavior of an organism [60]. The serotonergic (5-HTergic) system is involved in the regulation of a series of physiological activities, such as hormone secretion, learning, immune regulation, and emotion regulation [61,62]. Therefore, sulpiride and clozapine may have adverse effects on an octopus by interfering with the normal physiological functions of DA and 5-HT in the octopus brain tissue, thereby seriously threatening its health.

The expression levels of functional molecules in the 5-HTergic signaling system were abnormal, with abnormal transcription levels of DRD1, DRD2, DRD3, DAT, 5-HTR1, 5-HTR1B, 5-HTR2A, SERT, and MAO. Abnormal protein expression levels of DAT, SERT, 5-HTR1, 5-HTR1B, and 5-HTR2B were also observed. The 5-HTergic nerve endings release 5-HT, regulating midbrain DA neuron activity and DA release by acting on the 5-HTR2A receptor of DA neurons or 5-HTR2C receptors of γ-aminobutyric acid neurons. Sulpiride also indirectly induces the upregulation of 5-HT synthase Tryptophan hydroxylase gene transcription by inhibiting the DAergic system. The sensitivity of the DRD2 receptors of DAergic neurons to receptor antagonists is higher than that of the DRD2 receptors in the post-synaptic membrane. Sulpiride in the high-dose group antagonizes DRD2 receptors in the pre-synaptic membrane and binds to DRD2 receptors in the post-synaptic membrane, thus producing negative feedback on the activity of the DAergic system. The transcription levels of functional molecules in the DAergic and 5-HTergic systems are downregulated in neurons. Therefore, sulpiride may cause abnormal discharge or the excessive release of DA neurons by antagonizing the cell body of DA neurons or DRD2 receptors in the pre-synaptic membrane of nerve endings, leading to an abnormal increase in DA levels in synaptic clefts. This stimulates the body to regulate the DA levels by reducing DA synthesis, increasing DA transport and recovery in the synaptic cleft, and inhibiting the pro-release of 5-HT on DA. In addition, 5-HTR2A receptors distributed in DA neurons regulate the sensitivity of DA receptors by regulating the tonic release of DA and regulating and modifying the activity of phasic DA induced by action potentials. Meanwhile, the 5-HTR2A receptor is involved in the negative feedback regulation of 5-HTergic neuron activity and 5-HT release. Activation

of 5-HTR1 receptors in the pre-synaptic membrane (including 5-HTR1A, which regulates neuronal firing, and 5-HTR1B, which regulates 5-HT release) leads to the inhibition of neuronal activity and reduced transmitter release. Hence, clozapine antagonizes 5-HTR2A and DRD4 receptors, resulting in increased DA synthesis and compensatory upregulation of the DA receptor and transporter transcription and expression levels in octopus brain tissues, and increased 5-HT synthesis, transport, and receptor transcription and expression levels. Finally, in many cases, no dose-dependent changes were observed, which may be caused by individual differences.

## 5. Conclusions

In conclusion, the antipsychotic drugs sulpiride and clozapine mainly antagonize DRD2 and 5-HTR2A receptors in the octopus brain tissue, which manifest as abnormal transcription and expression levels of DAergic and 5-HTergic functional receptors, causing functional abnormalities in the DA and 5-HT neurotransmitter systems; they may interfere with the normal physiological functions and seriously threaten the health of octopuses. It is essential to note that there were some limitations to the present study. A typical example is that we were unable to ascertain whether the drug concentration is constant or whether other pollutants will amplify the efficacy of fluoxetine and sulpiride in seawater. Nonetheless, we believe that this study provides basic data for further study of the effects of antipsychotic drugs on important physiological functions of octopus brain tissue and promote the improvement of our understanding of ecological environmental behaviors.

**Author Contributions:** Conceptualization, X.P. and Q.X.; methodology, B.Z.; software, Q.X.; validation, B.Z., Y.G.; formal analysis, X.P.; investigation, X.P.; resources, B.Z.; data curation, Q.X.; writing—original draft preparation, X.P.; writing—review and editing, B.Z.; visualization, Y.G.; supervision, Y.G.; project administration, Y.G.; funding acquisition, B.Z. All authors have read and agreed to the published version of the manuscript.

**Funding:** This research was funded by Bo Zhang grant number LY19C030001 And The APC was funded by LY19C030001. This work was supported by the National Natural Science Foundation of Zhejiang Province, China (No. LY19C030001).

**Institutional Review Board Statement:** Not applicable.

**Informed Consent Statement:** Not applicable.

**Data Availability Statement:** The data supporting reported results can be acquired by contacting the corresponding author.

**Conflicts of Interest:** The authors declare that there is no conflict of interest.

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
