# Peer review of "Effects of Environmentally Relevant Concentrations of Antipsychotic Drugs (Sulpiride and Clozapine) on Serotonergic and Dopaminergic Neurotransmitter Systems in Octopus Brain Tissue"

_water, doi:10.3390/w14172608_

Round 1

Reviewer 1 Report

This manuscript deals with the important ecological issue of the effect of  residual drug pollution on octopuses. The authors selected 2 interesting drug candidates, which act on dopamine and 5HT systems, and exposed octopuses to varying doses over a period of several days. While the manuscript is interesting there are several major issues that need to be addressed.

The methods section does not provide adequate detail regarding the exposure of animals to the chemicals. In a study such as this, in which drug doses are added into the water the animals are kept in, the complete details of the water circulating system these drugs are being added to is essential. Additionally, there is no measurement or quantification of how much drug in effect stays in the water.

·       Another issue is the identity of the species of octopus used. The authors give no citation regarding this species Octopus ochellatus. As I assume it is not a misspelling, I can’t find any real sources in the literature regarding this species as it is spelled in the manuscript. Is this an alternative name to Octopus ocellatus (Amphioctopus fangsiao). This needs to be clarified, with citations to relevant literature and general information known about the species added to the introduction.

Another major issue, is the inconsistency in the he results section. Under the different headings I find that not all results marked as significant on the graphs are mentioned in the results. Those that are mentioned seem to written in a random order.

For example: in the results 3.1.1 and in the first figure (fig1) for the lowest concentration all the values that are upregulated are mentioned (but not in the order in which they appear in the graph).  In the 50ng/L concentration only DAT, SERT and 5-HTR1B are mentioned, but according to the graph there is a sig. difference in other values as well. For the highest concentration all the values that are downregulated are mentioned (but again, not in the order in which they appear in the graph).  The final sentence mentions DRD2 with no mention of dosage - the graph shows a significance only in the 50ng/L, which is strange in itself because all the values are extremely low.

This issue seems continuous throughout the results of the manuscript. It further makes the discussion very hard to follow and understand.

Finally, in many places in the manuscript doses are compared to each other, in these cases, a statistical test directly comparing the doses is required. A more or less significant difference between dose and control does not necessarily cover the difference between doses.

Minor issues in the introduction:

lines 66-80

This sections contains details about the sources and concentrations of sulpiride and clozapine in the environment. The numbers they quote are either input/output ranges of treatment plants or fresh water sources such as rivers. I think a connection should be made between these data and similar information about coastal or sea water levels of pharmaceuticals (such as in Bidel et al. 2016, Neurotoxicology), or a connection between waste sources and the sea pollution (such as in Gaw et al. 2014 Philos Trans R Soc Lond B Biol Sci.).

lines 82-120

The entire paragraph switches back and forth many times from cephalopods, to mullosks, to octopuses. There should be an order to the paragraph which allows the readers to understand what the known information about octopuses is vs. cephalopods or mullosks in general.

In general - This section needs to be rewritten and all the references rechecked. The references do not match the text. In some places the authors mention squid regarding a paper about cuttlefish, which are very different from squid.

Minor issues in the methods:

How were the animal collected?

Details about animal keeping are missing  - individual or group housed, water volume or tank size per animal.  

Details about disposal of sea water + drugs

Mention of the distribution of sex into the treatment group.

What does a 1% ethanol control mean.

Reviewer 2 Report

General comments:

Although this study proposes interesting data on drug contamination in water and emphasises the danger for populations of aquatic organisms, it needs revision before it can be positively evaluated for publication in 'water'. Before going into detail with the specific comments, I would also like to ask the authors why they chose not to perform a histological examination to assess the condition of brain tissue, as well as the evaluation of the expression of inflammatory markers and apoptotic cells.

Lines 51-65: No reference has been included to support this entire portion of the text. Please add some

Line 66: Which evidence? There are again no citations in support of this assertion

Lines 127-144: In the introduction section the authors provide references about environmental concentrations of Sulpiride and Clozapine. However, it’s not clear why the authors had selected these particular concentrations (20, 50, and 100 ng/L) above the several different concentrations of the same drugs detected in the aquatic environment. I suggest the authors to revise this section

Lines 136-141: The exact number of the experimental groups it’s not clear (10 octopuses for group, but how many groups?). I suggest the authors to take a cue from these papers and cite them https://doi.org/10.3390/toxics10020081, https://doi.org/10.3390/toxics10050272

Materials/Methods and results: I suggest that the authors, in order to improve the quality of the manuscript and the results presentation, take a cue from this work and cite it  https://doi.org/10.3390/life12010128  

Figure 4: This figure it’s not good enough, i suggest the authors to provide a better one.

Line 271: rewrite the sentence to show that every data is compared to the control group (it’s too much repeated)

Author Response

请看附件

Reviewer 3 Report

please find my comments directly in the manusript attached

Round 2

Reviewer 3 Report

thank you for considering my comments and suggestions